# Ecological forecasts for marine resource management during climate extremes

Stephanie Brodie [1,2,3] ✉, Mercedes Pozo Buil[1,2], Heather Welch[1,2], Steven J. Bograd [1,2], Elliott L. Hazen [1,2], Jarrod A. Santora[4,5], Rachel Seary[1,2,4], Isaac D. Schroeder[1,2] & Michael G. Jacox [1,2,6]

Forecasting weather has become commonplace, but as society faces novel and uncertain environmental conditions there is a critical need to forecast ecology. Forewarning of ecosystem conditions during climate extremes can support proactive decision-making, yet applications of ecological forecasts are still limited. We showcase the capacity for existing marine management tools to transition to a forecasting configuration and provide skilful ecological forecasts up to 12 months in advance. The management tools use ocean temperature anomalies to help mitigate whale entanglements and sea turtle bycatch, and we show that forecasts can forewarn of human-wildlife interactions caused by unprecedented climate extremes. We further show that regionally downscaled forecasts are not a necessity for ecological forecasting and can be less skilful than global forecasts if they have fewer ensemble members. Our results highlight capacity for ecological forecasts to be explored for regions without the infrastructure or capacity to regionally downscale, ultimately helping to improve marine resource management and climate adaptation globally.

Climate variability and change is altering the structure and function of ecosystems globally causing disruption and uncertainty to human and ecological communities[1–3]. There is a critical need for forward-looking information on ecosystem conditions to support resource management and decision-making. To meet this demand, sub-seasonal to decadal forecasting of atmospheric and ocean physics and ecology has rapidly advanced. There are now forecast products designed to support flexible management frameworks and resilient communities that are capable of responding to climate-driven change[4–7].

Ocean forecasting efforts are at the forefront of this new research agenda. A range of physical and ecological properties provides enhanced predictability of marine ecosystems for seasonal (e.g. ocean memory, life history traits)[8–10] to decadal[11–13] lead times. Forewarning of

ocean and ecosystem conditions at such lead times can support proactive decision-making for the blue economy[4,11,14,15]. Yet, uptake of these forecasts by end-users can be impeded by technical debt, inadequate representation and communication of uncertainty, and compatibility of a forecast with the needs of its targeted end-user[5,16–18]. Indeed, one of the barriers to forecast development is a common presumption that high-resolution regional ocean forecasts are required to develop marine forecasting applications. Yet, high resolution regional forecasts are often not readily available for both the hindcast and operational configurations required. Furthermore, studies have shown that global forecasts have some utility for regional ecological forecasting applications[7,19,20]. There is thus value in further understanding the limitations and applicability of readily available

[1]Institute of Marine Sciences, University of California Santa Cruz, Monterey, CA, USA. [2]Environmental Research Division, Southwest Fisheries Science Center, National Marine Fisheries Service, National Oceanic and Atmospheric Administration, Monterey, CA, USA. [3]Environment, Commonwealth Scientific and Industrial Research Organisation (CSIRO), Brisbane, Queensland, Australia. [4]Fisheries Ecology Division, Southwest Fisheries Science Center, National Marine Fisheries Service, National Oceanic and Atmospheric Administration, Santa Cruz, CA, USA. [5]Department of Applied Math, University of California, 1156 Santa Cruz, CA, USA. [6]Physical Sciences Laboratory, Earth System Research Laboratories, National Oceanic and Atmospheric Administration, Boulder, CO, USA. ✉e-mail: steph.brodie@csiro.au

operational global physical forecasts as compared to downscaled forecasts for regional applications.

Here, we use two case studies to explore the potential for transitioning existing management tools from nowcast application to forecasting application using downscaled (~10 km resolution) and global (~100 km resolution) model forecasts with 0.5-11.5 months lead time. The tools are: (1) the Habitat Compression Index (HCI; Fig. 1)[21,22], which identifies when cool thermal habitat area used by whales is compressed nearshore, increasing entanglement risk in a fixed gear crab fishery; and (2) the Temperature Observations to Avoid Loggerheads tool (TOTAL, Fig. 1)[23], designed to guide the timing of a drift gillnet fishery closure to reduce bycatch risk of loggerhead sea turtles (*Caretta caretta*). Both tools use sea surface temperatures and are based in the California Current Ecosystem (CCE) - a highly productive ecosystem in the Northeast Pacific that is characterized by seasonal upwelling of cool, nutrient-rich waters[24,25], and for which skilful forecasts of sea surface temperatures (SST) are possible with lead times of several months[26]. The CCE is a critical foraging ground for culturally, socially, and economically important protected species that are under threat from diverse anthropogenic activities[27]. We show how forecast configurations of these two management tools can be skilfully forecast up to 12 months in advance and are capable of forewarning human-wildlife interactions caused by unprecedented climate extremes. Advanced warning of potential threats to these species is key to developing proactive rather than reactive management strategies that can help to reduce uncertainty and anxiety in the face

of novel ocean conditions and novel resource management challenges.

## Results
### Skilful global forecasts during climate extremes

Global forecasts of SST can be applied to two resource management tools to skilfully forecast ocean conditions up to 11.5 months in advance. For the HCI, forecasts of high compression events – which are associated with high whale entanglement risk – were typically skilful from 0.5 to 1.5 months lead time across all months of the year (Fig. 2B–D). Significant forecast skill of high compression events extends out to 8.5 months for forecasts in February-March. Overall we see that forecast skill varies depending on the metric of skill used (Fig. 2B–D) but in general skill is higher in winter and spring, which are important seasons for mitigating whale entanglement risk as they coincide with both fishing activity and whale migrations (Fig. 2B–D). If an advanced warning system was in place (i.e., an operational HCI forecast), there could have been 0.5-11.5 months warning of historical high compression events. For example, high habitat compression was observed during May 2005, and was correctly predicted 0.5, 1.5, and 3.5 months in advance. In another example, continuous high habitat compression was observed during a large marine heatwave (Mar 2014–Dec 2016) and 94% of our forecasts ($n = 193$) correctly identified high habitat compression over this 33-month period (Fig. 2A). These HCI forecasts correctly predicted high compression up to 11.5 months in advance, with most false negatives (where high compression was not predicted) occurring during the onset of the marine heatwave in early 2014 (Fig. 2).

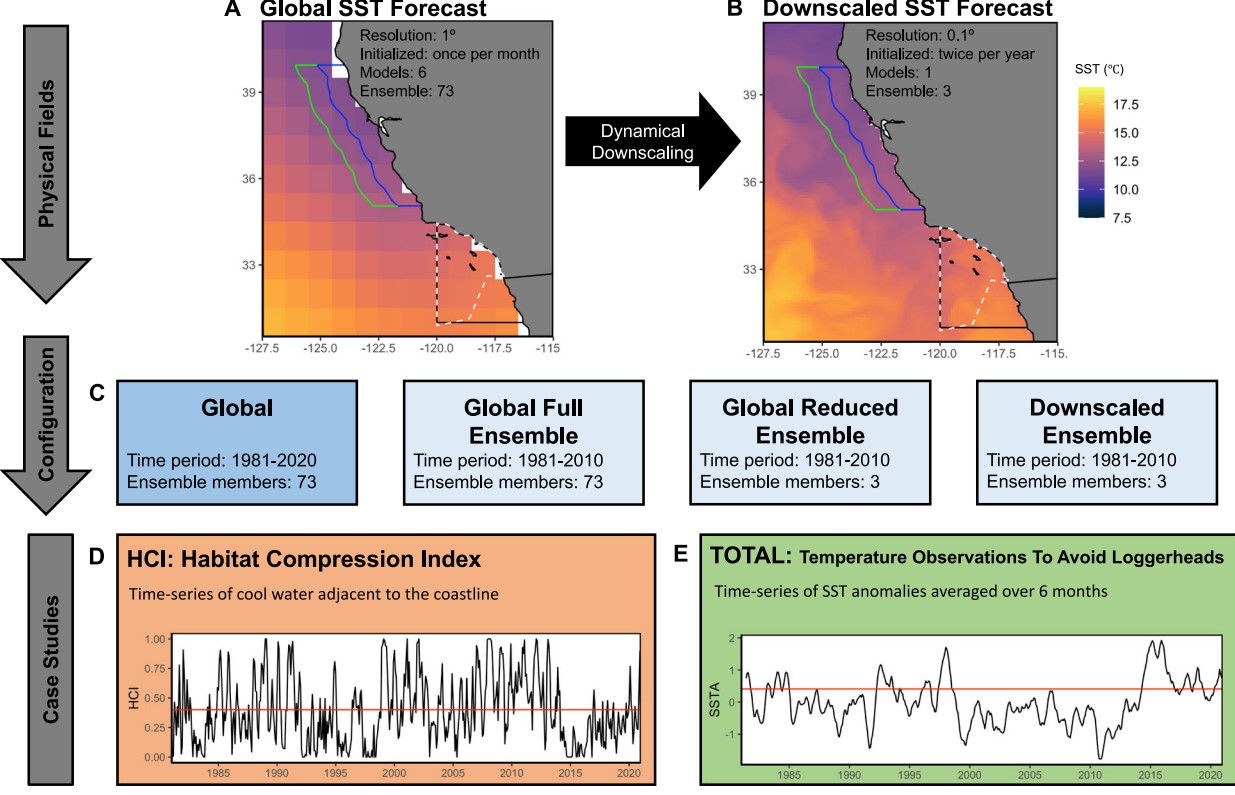

**Fig. 1 | Conceptual outline of the study highlighting forecast fields and case studies. A** Global SST forecasts and (**B**) downscaled SST forecasts were used in (**C**) four configurations for two management tools to test how the spatial resolution and ensemble size affect forecast skill. The two management tools are the (**D**) Habitat Compression Index (HCI), and (**E**) Temperature Observations to Avoid Loggerheads (TOTAL). Maps of SST forecasts represent ensemble member 2 of the CanCM4 models at lead time 0.5 for 2010-01-01. Black box on SST maps represents the spatial domain of the TOTAL tool, and the white dashed line the loggerhead turtle closure. Blue and green lines represent the domain (75 km and 150 km from shore, respectively) of the HCI tool. For the case studies, the red line on HCI indicates the long-term mean with values below this considered as high habitat compression; and the red line on TOTAL indicates the threshold to enact a potential closure. Source data are provided as a Source Data file.

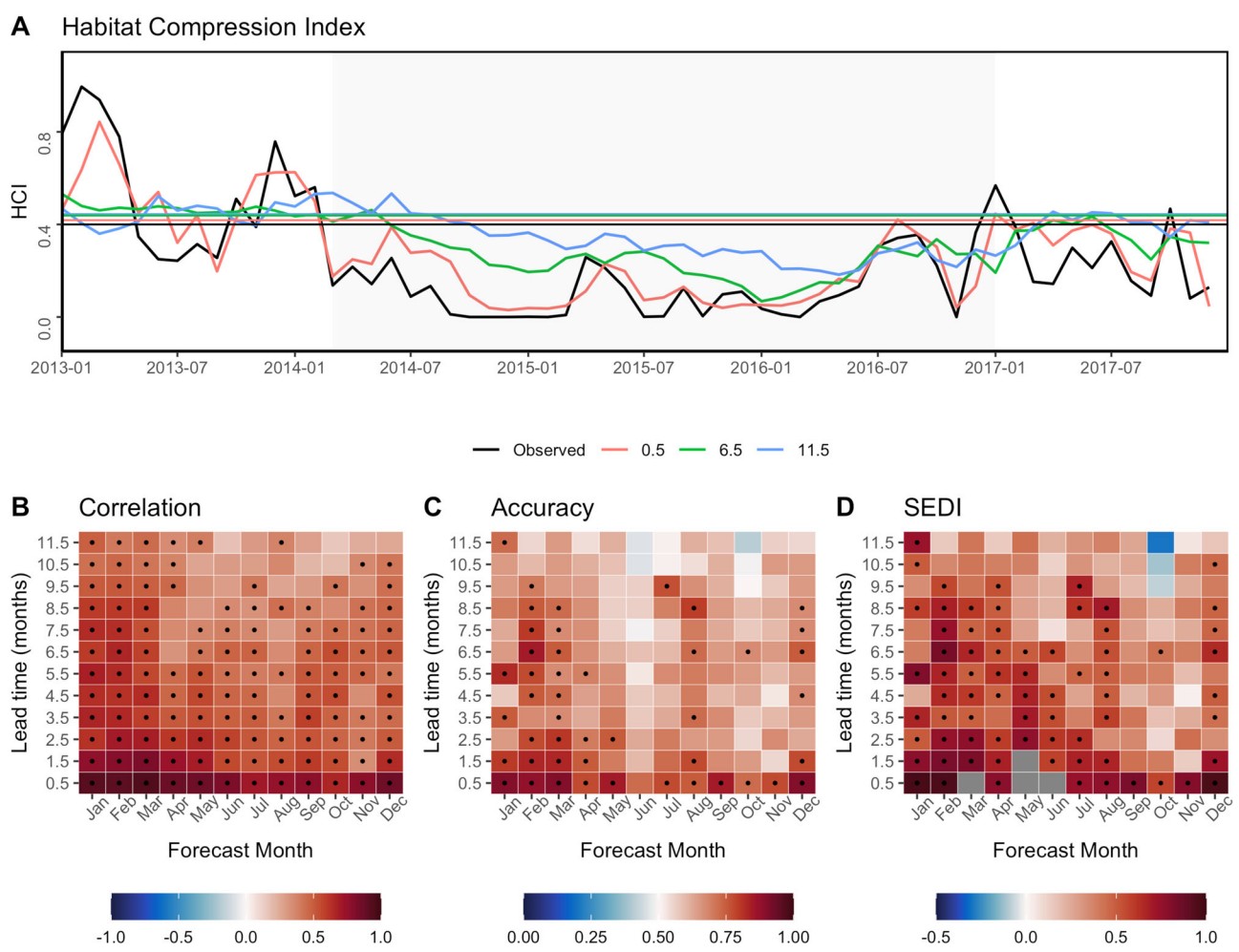

**Fig. 2 | Skilful forecasts of the Habitat Compression Index.** Observed and forecast Habitat Compression Index using global sea surface temperature forecasts, and its associated skill assessment. **A** Time-series of observed (black) and forecast Habitat Compression Index (HCI) at three example lead times (0.5, 6.5, 11.5 months corresponding to red, green, and blue colors) with gray shading indicating high compression during a marine heatwave (Mar-2014 to Dec-2016). Horizontal lines indicate the long-term mean of HCI, with high compression as any values below the long-term mean, with colors matching their corresponding forecast lead time. Forecast HCI is the ensemble mean of 73 global forecast models. **B**–**D** Skill assessment of forecast HCI for each target month and each lead time, showing the correlation coefficient, forecast accuracy, and SEDI (Symmetric Extremal Dependence Index) calculated across all years (1981–2020). For correlation values, black dots indicate forecast skill is significantly greater than zero (95% confidence). Accuracy values > 0.5 and SEDI values > 0 indicate the forecast is better than random chance, with black dots indicating skill is significantly greater than random forecasts (95% confidence). Gray SEDI squares are months when there are no false positives and SEDI can't be computed. Source data are provided as a Source Data file.

In our second case study, TOTAL closure conditions were skilfully forecast at 6.5 months lead time, with significant skill extending out to 11.5 months by some metrics (Fig. 3). TOTAL is the average of temperature anomalies across the 6 months preceding the target month (June, July, or August), with closures recommended when anomalies exceed a threshold. If an operational forecast system was in place for TOTAL, we could have had up to 11.5 months warning of potential closures. For example, a closure was enacted in August 2014 and our TOTAL forecast correctly predicted the closure as early as November 2013 (Fig. 3). During the marine heatwave of 2014-2016, TOTAL closures were enacted in Jun-Aug 2015 and 2016, and our retrospective TOTAL forecasts correctly predicted all these closures 11.5 months in advance (Fig. 3).

### Downscaled versus global forecasts
Comparing the performance of downscaled versus global forecasts identified advantages and disadvantages of downscaling for these short-term forecasts, and highlighted the utility of global forecasts despite their coarse spatial resolution. For our two case studies, the full global forecast ensemble was typically more skilful than the

downscaled ensemble. The increased skill arises from the greater number of ensemble members available (73 versus 3 ensemble members; Fig. 4; S1, S2). Forecast skill of the global models based on only 3 ensemble members showed a dramatic reduction and performed worse than the downscaled forecasts (Fig. 4; S1; S2). Thus, while downscaling does appear to improve the skill of individual ensemble members, the lower skill of global ensemble members can be offset by the availability of a greater number of ensemble members that better characterize environmental variability.

### Discussion
We show that two resource management tools configured to forecast can provide accurate forward-looking information 0.5-11.5 months in advance. Our approach demonstrates the value of operational forecast systems to support decision-making for ocean end-users faced with uncertain and variable future conditions. Importantly for both tools, accurate predictions could be made using readily available global forecasts even though they have relatively coarse resolution. While regional downscaling did offer value for increasing forecast skill (especially for the TOTAL forecast), the improvements afforded by

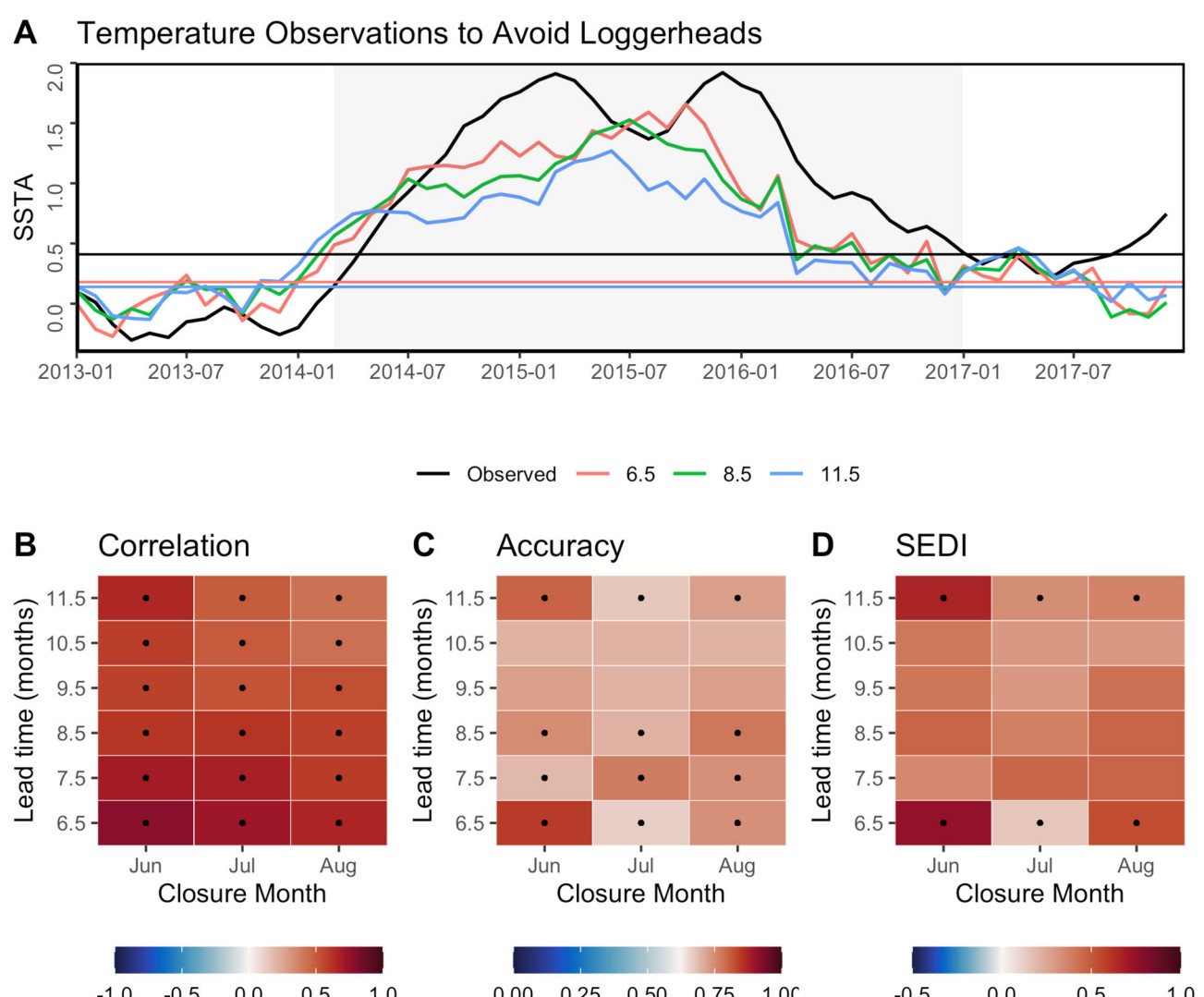

**Fig. 3 | Skilful forecasts of the sea turtle bycatch closures using TOTAL (Temperature Observations to Avoid Loggerheads). A** Time-series of observed (black) and forecast (color) sea surface temperature anomalies (SSTA) averaged by 6-months at three example lead times (6.5, 8.5, 11.5 months corresponding to red, green, and blue colors). Gray shading indicates a marine heatwave from Mar-2014 to Dec-2016. Straight lines indicate the 74% lead-time specific quantile threshold to enact a closure in either June, July, or August, with colors matching their corresponding forecast lead time. Forecast SSTA is the ensemble mean of 73 global forecast models. **B–D** Skill assessment of forecast TOTAL for each closure month and each lead time, showing the correlation coefficient, forecast accuracy, and SEDI (Symmetric Extremal Dependence Index) calculated across all years (1981–2020). For correlation values, black dots indicate forecast skill is significantly greater than zero (95% confidence). Accuracy values > 0.62, and SEDI values > 0 indicate the forecast is better than random chance. Source data are provided as a Source Data file.

downscaling were overshadowed by the ability of global forecasts to use a much larger number of ensemble members from global forecasts. The results of our study help to lower barriers to implementing marine ecological forecasting tools in the California Current and other regions, as we demonstrate that researchers and practitioners can work with readily available global forecasts for some applications without the need for costly downscaling. However, we discuss below the pros and cons of global and regionally downscaled forecasts, with the aim of helping practitioners decide whether to invest in regional downscaling for marine ecological forecasts. Researchers should test the performance of available forecasts before assuming their resolution precludes a specific application. Providing demonstrable examples of marine ecological forecasting applications with open-access forecasts will help to increase accessibility and capacity to advance the field globally.

Climate extremes interacting with longer-term climate change are leading to unprecedented environmental conditions globally[28]. To adapt to these uncertain and variable conditions, there is a need to

develop tools that provide seasonal to annual information on future ocean conditions[29]. Here, we demonstrate that forecast configurations of two existing management tools have the capacity to skilfully predict highly variable SST ocean conditions and provide advanced warning for a prolonged marine heatwave that had an unprecedented number of whale entanglements and loggerhead turtle bycatch events. For the HCI, forecasts of high compression events could be used within the current decision framework (the RAMP: Risk Assessment and Mitigation Program) that meets monthly to guide fishing closures[30,31]. These monthly meetings consider historical and current conditions in the CCE, but an operational HCI forecast system would allow the RAMP to integrate the likelihood of future compression events when deciding on entanglement mitigation actions (e.g. gear usage, fishery closures, fishing season delays; https://www.opc.ca.gov/risk-assessment-and-mitigation-program-ramp/)[30]. Importantly, an operational HCI forecast would allow more proactive management (i.e., a closure with advanced warning because of high-risk conditions) rather than reactive management (i.e., closures following an entanglement event). The

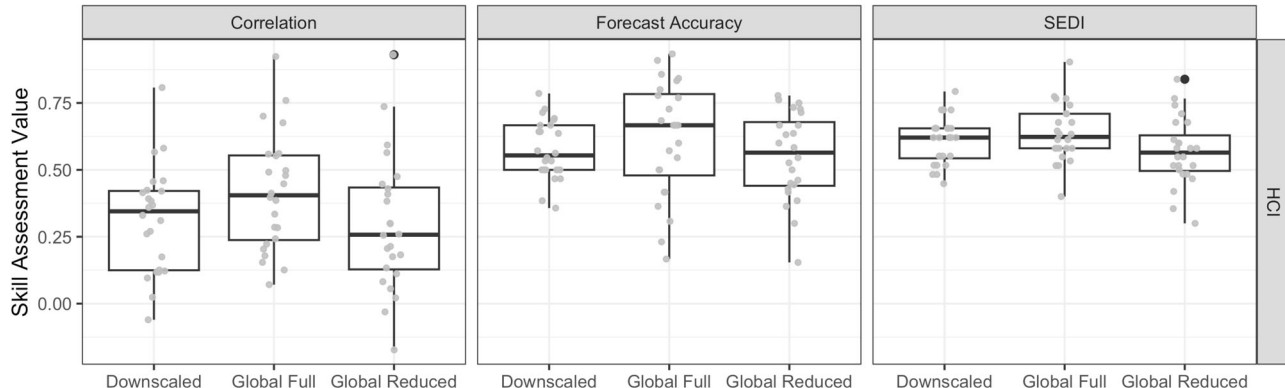

**Fig. 4 | Improved skill from increasing number of ensemble members.** Skill assessment comparison among three forecast configurations (Downscaled Ensemble with 3 ensemble members, Global Full Ensemble with 73 ensemble members, and Global Reduced Ensemble with 3 ensemble members) for the Habitat Compression Index (HCI) tool. Skill assessments include correlation coefficients, forecast accuracy, and SEDI. Values are for each month and lead time from 1981 to 2010 for the HCI tool ($n = 24$ forecasts). We only show global forecasts that were initialized in January and July to facilitate comparisons with the downscaled forecasts. Values for Temperature Observations To Avoid Loggerheads (TOTAL) are not shown as too few forecasts were available for boxplot comparison ($n = 3$), but see Fig. S2. Boxplots show the median as a solid black line, the lower and upper hinges are the first and third quartiles, whiskers extend from the smallest/largest value no further than the 1.5*interquartile range, with outliers (black circles) beyond the edge of whiskers plotted individually. Source data are provided as a Source Data file.

consequences of these mitigation actions are steep, and decision makers must navigate trade-offs between the conservation of species legally protected under the U.S. Marine Mammal Protection Act, and the socio-economic importance of California's most lucrative fishery[30–32]. Important next steps would be to begin co-developing an operational HCI forecast with the RAMP end-users to provide critical insight into how a product could be produced for improved use and uptake[5]. Ecosystem-based management could be advanced by improved integration of forecasts of ecosystem conditions[33,34]. Thus, the value of an operational HCI forecast could be a linchpin towards developing climate-ready ecosystem-based management approaches in the face of increased climate impacts[33,35].

Likewise, the TOTAL tool has potential to act as an ecosystem indicator for the Southern California Bight. TOTAL was designed to indicate increased likelihood of loggerhead presence to guide the timing of fishery closures to reduce turtle bycatch in the Drift Gillnet fishery. However, due to a dramatic reduction of fishery effort in recent years[36] there may be no need to support an operational configuration of this tool. However, our demonstration of good skill for the TOTAL forecast may lead to increased capacity to modify the tool to support alternative resource management decisions. For example, the rare occurrence of loggerheads in the Southern California Bight has made it challenging to create observational and sampling programs, and a TOTAL forecast indicating increased likelihood of loggerhead turtle presence would provide the necessary lead time to prepare and conduct required monitoring.

Regional downscaling of physical and biogeochemical fields can resolve fine-scale features in response to large-scale forcing from global climate models[8,37]. Our results show how downscaling can improve the skill of individual ensemble members, but that using a greater number of ensemble members from global forecasts has the capacity to improve skill beyond that of downscaled forecasts. We attribute this improvement in skill to an increased characterization of environmental variability derived from the greater number of ensemble members. Our results indicate that the best approach to maximize skill for ecological forecasting applications would be to regionally downscale all available ensemble members. However, this is not logistically or computationally feasible, which raises the question of how best to determine whether to invest in regional downscaling for marine ecological forecasts. We have developed a table to support practitioners faced with this question (Fig. 5) and highlight the need to

consider the environmental fields required, the spatiotemporal scale of the tools, and the accessibility of knowledge required, in order to determine whether downscaling is necessary for a particular application or region (Fig. 5). For example, the coarse spatial and temporal resolution of the two management tools doesn't require the 0.1° resolution of the regionally downscaled fields. Tools that require more finely resolved fields (<1° resolution), or for regions where fine-scale physical processes need to be captured (e.g. tides or shelf-processes), will likely benefit from downscaled forecasts, and tests should be performed in those cases to ensure downscaling is providing added value (Fig. 5) (e.g. ref. 38). We only explored SST as a variable but if biogeochemical fields are needed for ecological forecasts (e.g. oxygen), then downscaling may be needed to ensure coupling between the required physical and biogeochemical models[9,15,34]. We recognize that our case studies are limited to the CCS, which is characterized by a relatively narrow shelf and exposed to basin-scale dynamics that confer greater predictability[26]. However, SST anomalies can be skillfully forecast in many coastal ecosystems worldwide, including for shelf-seas[10]; in those systems we suggest that researchers first test the performance of available global forecasts before assuming their resolution precludes application.

The field of ecological forecasting is still in its infancy, and with increasing social-ecological costs of climate change there is a critical need to provide forward looking information to conservation practitioners and resource managers faced with increased uncertainty about the future. We demonstrate the capacity to configure two applied resource management tools to forecasting systems with the ability to forewarn changing ecological conditions caused by unprecedented climate extremes. These successful case studies demonstrate that transitioning regional operational tools to forecasts can be logistically and economically feasible by relying on open-access operational global forecasts. Importantly, our results highlight capacity for ecological forecasting applications to be explored for nations and regions without the infrastructure or capacity to regionally downscale (e.g., developing nations), ultimately helping to improve marine resource management and climate adaptation globally.

## Methods
### Summary
We configure two existing resource management tools, originally configured to use observed (historical) ocean temperatures, to a

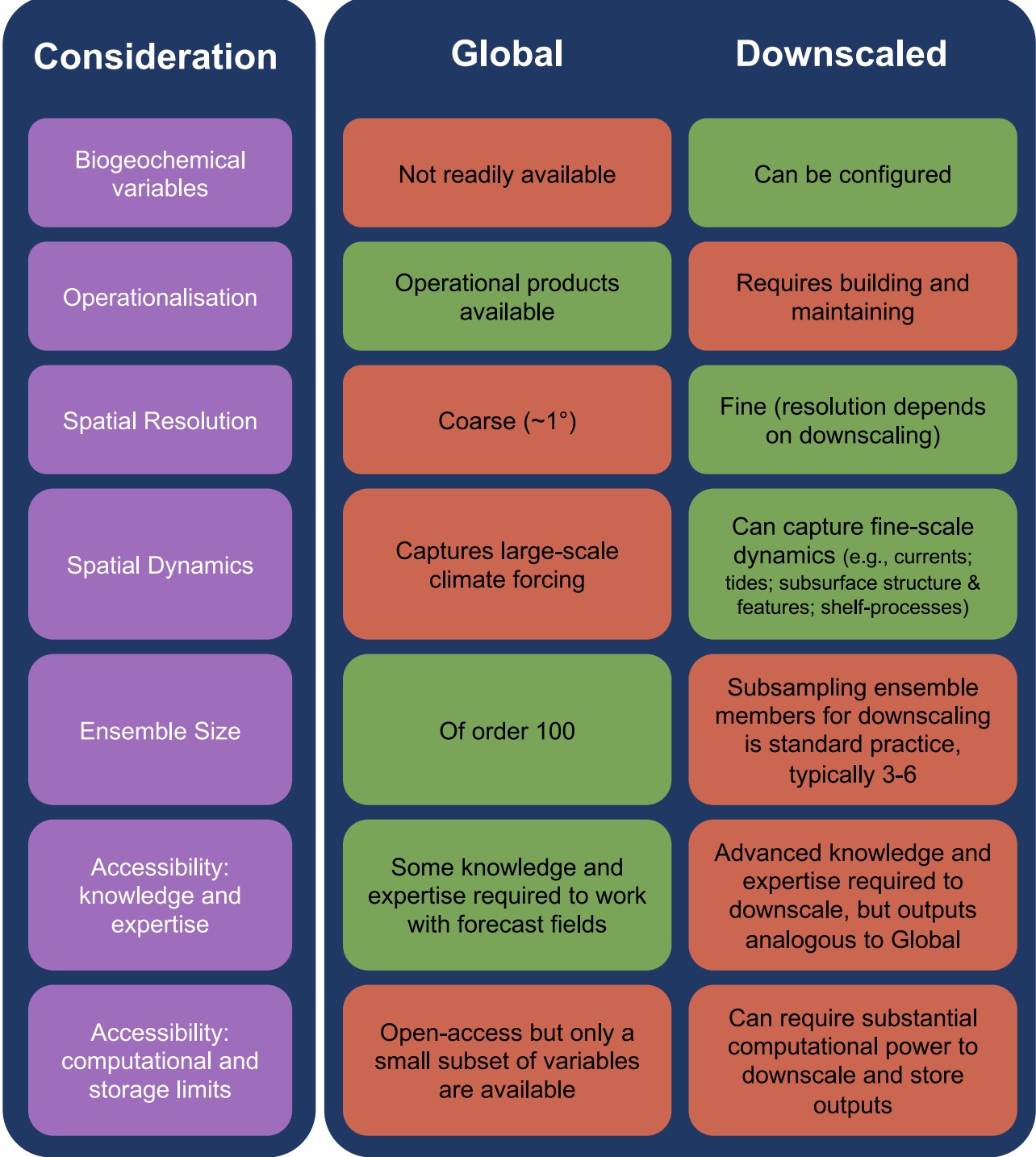

**Fig. 5 | Considerations to support practitioners deciding whether to invest in regional downscaling for marine ecological forecasts.** For global and downscaled forecasts, colors indicate a dichotomy of superior (green) compared to inferior (red).

forecasting system and conduct a retrospective forecast to test their skill. We first conducted a retrospective forecast using global forecasts (73 ensemble members) across the full historically available period (1981–2020) – termed the Global model. We then compared the performance of three forecast configurations: First, we used global forecasts (73 ensemble members) across a reduced historical period (1981–2010) – termed the Global Full Ensemble. Second, we used forecasts regionally downscaled (3 ensemble members) to the CCE for the same reduced historical period (1981–2010) – termed the Downscaled Ensemble. Third, we used a reduced subset of the global forecasts (3 ensemble members) for the same reduced historical period (1981–2010) – termed the Global Reduced Ensemble. All forecasts are compared to SST observations, extracted from a CCE regional reanalysis[39]. This reanalysis is based on the Regional Ocean Modeling System (ROMS) and covers the west coast of the U.S. (30-48˚N, 134-115.5˚W) with 0.1 degree (~10 km) horizontal resolution and 42 terrain-following vertical levels[40].

## Global forecasts

Global forecasts of monthly sea surface temperature were obtained from the North American Multimodel Ensemble (NMME; Table S1; https://www.cpc.ncep.noaa.gov/products/NMME/)[41]. We selected six global forecast models from the NMME (Table S1), which are initialized every month and, depending on the model, predict 9–12 months into the future at a monthly resolution. These six models were chosen as they have multidecadal retrospective forecasts with which we can assess skill and are currently active in the NMME. Each global model has up to 24 ensemble members (Table S1), with each ensemble member starting from slightly different initial conditions to characterize natural climate variability. We used forecasts from a total of 73 ensemble members (Table S1) across the six models. Forecast spatial resolution is 1°, with forecasts available from 1981 to present day (exact date varies among models; Table S1). Three of these 73 ensemble members were chosen for the Downscaled Ensemble and Global Reduced Ensemble configurations (see below).

## Regional forecasts

A subset of forecasts was dynamically downscaled using ROMS with the 0.1° configuration described above for the CCE regional reanalysis. Due to computational limitations of producing and storing downscaled forecasts, the suite of ensemble members and initializations had to be reduced relative to what is available for global forecasts. The global forecast model chosen to be downscaled was the fourth version of the Canadian Center for Climate Modeling Analysis' coupled climate model (CanCM4i; Table S1)[42]. CanCM4i has been shown to produce skilful SST forecasts in the CCE[26]. Three ensemble members (2, 8, and 10) from the CanCM4i output were downscaled (chosen to capture the approximate spread in skill of all 10 ensemble members); and forecasts were initialized twice per year - once in summer (July) and once in winter (January). For each forecast, atmospheric and oceanic output was obtained from each single ensemble member of the global model and was bias corrected before being used to force ROMS[43]. Specifically, global forecast anomalies were calculated relative to their own climatology, and then the global forecast anomalies were added to observed historical climatologies derived from high-resolution atmospheric and oceanic reanalyses. These bias corrected fields were then used to force ROMS at the surface and lateral ocean boundaries. High-resolution climatologies were extracted from the European Center for Medium-Range Weather Forecasts version 5 atmospheric reanalysis (ERA-5)[44] at hour resolution and from the Simple Ocean Data Assimilation version 2.1.6 ocean reanalysis (SODA)[45] at monthly resolution. Downscaled forecast spatial resolution is 0.1°, with forecasts available from 1981-2010 at daily resolution.

## Case study 1: Habitat Compression Index

The Habitat Compression Index (HCI) is a regionally resolved measure of cool thermal habitat along the U.S. West Coast; the index presented here monitors surface water conditions off California (35–40°N). The HCI is used to assess the degree to which upwelling habitat (indicated by cool water) is compressed against the coast, as nutrient-rich upwelled waters attract whales seeking enhanced foraging opportunities. When upwelling habitat is compressed against the coast, it can increase overlap between whales and human activities, leading to entanglement in fishing gear[21,22,46]. Further, the HCI relates to the distribution and abundance of anchovy and other coastal pelagic species, providing inference on potential ecosystem shifts in the forage base[22]. The HCI was calculated as the number of grid cells with SST lower than a monthly SST threshold within 150 km of the coastline (Fig. 1 green line). The HCI was normalized by the total number of grid cells of the 150 km domain to scale values from 0 to 1. Monthly SST thresholds are the mean monthly SST from 1981-2010 from the coast to 75 km offshore (Fig. 1 blue line). Low

HCI values represent high compression, or reduction of cool thermal habitat, and are the primary interest to resource managers tasked with mitigating whale entanglement risk (Fig. 1). The long-term mean of the HCI is used to identify a high compression event (i.e. values below the mean)[21,22].

## Case study 2: TOTAL tool

The Temperature Observations to Avoid Loggerheads (TOTAL) tool monitors anomalously high SST in the Southern California Bight (31-34°N, 120-116°W) as an indicator of turtle bycatch risk and to recommend potential implementation of a fishery closure[23]. In response to historical loggerhead bycatch events in the California drift gillnet fishery during warm water years, the Loggerhead Conservation Area was created – which is a spatial closure enacted during months when water temperatures are anomalously warm (Fig. 1). TOTAL was calculated as the 6-month rolling mean of SST anomalies in the Southern California Bight domain. The spatial closure is potentially enacted during three months of the year (June, July, August) based on SSTA of the preceding six months. If SSTA exceeds a threshold, calculated as the minimum monthly anomaly value preceding three historical closure periods (Aug 2014, Jun-Aug 2015, & Jun-Aug 2016), a closure is recommended[23].

## Forecast comparison

We conducted a retrospective forecast for the Global model (73 ensemble members for 1981–2020), as well as for the three forecast configurations that aimed to compare the performance of global and downscaled forecasts (Fig. 1; 1981–2010).

The HCI was forecast for 0.5–11.5 month lead times, starting from each forecast initialization, using forecast SST. The HCI was calculated for each of the global ($n = 73$) and downscaled ($n = 3$) ensemble members, and then individual members were averaged together to create ensemble mean HCI forecasts for the different forecast configurations. The monthly SST thresholds (mean monthly SST from the coast to 75 km offshore) used to calculate the HCI were based on years 1981–2010. Monthly SST thresholds were specific to each forecast ensemble member and to each lead-time to account for model drift[10]. A lead-time specific long-term mean of the HCI was used to identify a high compression event (i.e., HCI values below the long-term mean).

For TOTAL, mean SSTA of the 6 months preceding potential closure months (Jun, Jul, Aug) was forecast. For global models, this resulted in forecasts with up to 11.5 months lead time (i.e., TOTAL used SSTA from lead months 0.5–5.5, 1.5–6.5,..., 5.5–11.5). For downscaled models, only two initialization dates were run (January and July) which limits the number of forecasts available (Table S2). That is, June closure estimates were the mean SSTA from Dec to May (lead 5.5 from July initialization and lead 0.5–4.5 from January initialization). July closure estimates were the mean SSTA from Jan to Jun (lead 0.5–5.5 from January initialization). August closure estimates were the mean SSTA from Feb-July (lead 1.5–6.5 from January initialization). TOTAL was calculated for each of the global ($n = 73$) and downscaled ($n = 3$) ensemble members, which were then averaged together to create ensemble mean forecasts. The monthly climatology used to calculate SSTA for TOTAL varied between configurations, where 1981-2020 was used for the Global model, and 1981-2010 was used for the remaining three configurations (Global Full Ensemble, Global Reduced Ensemble, and Downscaled Ensemble; Fig. 1). The threshold to identify a potential closure was calculated at each lead time, based on the forecast ensemble mean.

## Skill assessment

Forecast skill of each management tool was assessed by comparing observed and forecast values using three metrics: (1) correlation coefficient, which provides a statistical measure of the strength of a

linear relationship between observed and forecast values; (2) forecast accuracy, which indicates the fraction of correct forecasts; and (3) the Symmetric Extremal Dependence Index (SEDI) which has several properties that make it well suited to quantifying skill for rare events[7,47]. Details and equations for metrics are described below. Each skill metric was calculated across forecast years ($n = 41$ for the Global model; $n = 31$ for Global Full, Global Reduced, and Downscaled Ensemble). Pearson correlation coefficients (two-sided) were calculated from observed and forecast HCI values, and in the case of TOTAL, observed and forecast SSTA. The significance of correlation coefficients were assessed after first accounting for sample autocorrelation by calculating the effective degrees of freedom ($N_{eff}$):[48]

$$N_{eff} = \frac{N}{\sum_{t=0}^{N-1} \left(1 - \frac{t}{N}\right) r_t^F r_t^O} \quad (1)$$

Where $N$ is the number of samples in the forecast timeseries, and $r_t^F$ and $r_t^O$ are autocorrelation coefficients for the forecast ($F$) and observed ($O$) timeseries at lag $t$. We then use the CorCI function in the DescTools R package[49] to calculate 95% confidence intervals for the correlation coefficients based on a Fishers Z transformation, which follows a $z$ distribution with $N_{eff}$ -3 degrees of freedom[10]. If the lower confidence interval is greater than zero, then forecast skill is significant.

The remaining two metrics, forecast accuracy and SEDI, use a categorical assignment to a contingency table for skill evaluation. That is, the classification of whether an event did or didn't occur, and whether we did or didn't forecast that event. For HCI, we categorize a high compression event as any forecast HCI below or equal to the lead-dependent long-term mean HCI (1981–2010 or 1981–2020). This calculation of the HCI is consistent with the existing management tool for assessing whale entanglement risk in the California Dungeness crab fishery[21,22]. For TOTAL, we categorize warm events as those that exceed a threshold based on the minimum monthly anomaly value preceding three observed historical closure periods (Aug 2014, Jun–Aug 2015, & Jun–Aug 2016)[23]. This calculation of the TOTAL threshold is informed by sea turtle sightings data and is consistent with the existing management tool for assessing bycatch risk in the Swordfish drift gillnet fishery[23]. The observed SSTA threshold is 0.41, which is the 82nd percentile for observed SSTA from 1981 to 2010, and 74th percentile for observed SSTA from 1981 to 2020. We converted this absolute value of SSTA to a percentile to facilitate TOTAL comparisons among forecasts.

Forecast accuracy is the sum of true positives and true negatives, divided by the total number of forecasts. In other words, it is the fraction of forecasts that are correct. Forecast accuracy varies between 0 and 1, with 1 being perfect. Forecast accuracy can be strongly influenced by the frequency at which the event being forecast occurs ($f$). For reference, the expected forecast accuracy for a randomly generated forecast is:

$$\text{FA}_{\text{rand}} = f^2 + (1-f)^2 \quad (2)$$

For HCI, high compression occurs approximately 50% of the time ($f = 0.5$), so forecast accuracy above 0.5 indicates skill is better than random chance. For TOTAL, with the SSTA percentile for closure set at 74% ($f = 0.26$, used for the Global model), forecast accuracy above 0.62 indicates skill is better than random chance. For the Global Full, Global Reduced, and Downscaled forecast ensembles, the SSTA threshold is set at 82% ($f = 0.18$), so skill above 0.71 is better than random.

SEDI is a useful metric for assessing the skill of forecasting extreme events[7,47]. It is non-degenerate, meaning that it does not trend towards zero or infinity as rarity increases; it is not influenced by changes in the frequency of events (known as base-rate independence); and it is equitable, meaning that all forecasts including random

forecasts receive the same expected score (zero), irrespective of what method is used to generate random forecasts[50]. SEDI is calculated as:

$$\text{SEDI} = \frac{\log F - \log H - \log(1-F) + \log(1-H)}{\log F + \log H + \log(1-F) + \log(1-H)} \quad (3)$$

Where $H$ is the hit rate (ratio of true positives to total observed events) and $F$ is the false alarm rate (ratio of false positives to total observed non-events), and was calculated using the verify function in the verification R package (v1.42)[51]. The highest SEDI score is one and scores above zero indicate forecasts better than random chance.

The significance of two skill metrics, forecast accuracy and SEDI, were quantified using bootstrapping. For every month and lead time, we sampled random forecasts (as either a 1 or 0 to classify whether an event did or didn't occur) from the observed events for every year and calculated forecast accuracy and SEDI. This was repeated 1000 times. The 95% confidence intervals were calculated from the skill of the 1000 random forecasts, where significance was defined as skill exceeding the 97.5 percentile of the random forecasts.

## Reporting summary

Further information on research design is available in the Nature Portfolio Reporting Summary linked to this article.

## Data availability

Global SST forecasts can be accessed at the North American Multi-Model Ensemble (http://iridl.ldeo.columbia.edu/SOURCES/.Models/.NMME/). Output from the UCSC CCS reanalysis is available from https://oceanmodeling.ucsc.edu. The downscaled SST forecasts and forecasts of HCI and TOTAL generated in this study have been deposited in dryad[52]. Source data are provided with this paper. The real-time HCI (https://oceanview.pfeg.noaa.gov/whale_indices/) and TOTAL (https://coastwatch.pfeg.noaa.gov/loggerheads/index.html) management tools are publicly accessible. Source data are provided with this paper.

## Code availability

Code used to make forecasts and figures is available on GitHub[53].

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

## Acknowledgements

Funding was provided by the NOAA Climate Program Office Modeling, Analysis, Prediction, and Projections program (NA17OAR4310108; MGJ), the California Current Integrated Ecosystem Assessment (no grant number; ELH), and the California Ocean Protection Council (2021-242-UCSC; RS). We thank Emily Becker for facilitating data access, and Desiree Tommasi for revisions to an earlier version of the manuscript. The NMME project and data dissemination is supported by NOAA, NSF, NASA, and DOE, and the NMME archive receives help from NCP, IRI, and NCAR personnel.

## Author contributions

S.B. and M.G.J. conceived the study. M.G.J. obtained the global SST forecasts and produced the bias corrected forcing for the regional downscaling. M.P.B. conducted the regional downscaling. S.B. conducted the analysis and wrote the first draft. S.B., M.P.B., H.W., S.J.B., E.L.H., J.A.S., R.S., I.S., and M.G.J. all contributed to the study design, interpretation and presentation of results, and writing and editing the manuscript.

## Competing interests

The authors declare no competing interests.
