## [Peer Review File · Nature Communications]

Ecological forecasts for marine resource management during climate extremesREVIEWER COMMENTS

Reviewer #1 (Remarks to the Author):

At a high level, this manuscript was methodologically sound, original, and provides a useful contribution to the emerging scientific literature on ecological forecasting.

My primary comments on the paper are a few small places where I think methodological details and figures could be made more clear:

* The paragraph on lines 291-298 is repetitive with lines 208-221.

* On line 304 it's not clear how Monthly SST thresholds were set.

* On lines 329-330 the explanation of forecast accuracy and SEDI are too short and it's not clear that equations for this will be presented later. Even when you get to it later, it's not clear what the "several properties" are that make SEDI useful.

* Line 348, the definition of a HCI event as any one that's below average seems like a really lenient threshold, and it's not clear how this connect to the species biology or historically observed bycatch.

* Line 352: also not completely clear how these thresholds were set. They do seem connected to historically observed events, but its not clear why these specific thresholds were chosen (e.g. was there some sort of logistic regression between SSTA and events?)

* Line 538: I'm not a fan of "available upon request" because there's a lot of evidence that such data quickly becomes lost to the community. Is there a reason HCI and TOTAL in particular can't be provided?

* Figure 2: the use of dots in 2B to indicate significance creates the impression in 2C and 2D that no values are significant. Furthermore, the threshold for significance is not 100% clear for these metrics -- I wonder if you could use some sort of Monte Carlo approach (e.g.,

bootstrapping) to generate a significance test?

* Figure 3, S1, S2: same. Would be nice to have a more clear visual indicator of significance

* Totally nit-picky, but I found the British spelling of "skilful" instead of "skillful" to be really distracting

Reviewer #2 (Remarks to the Author):

The authors show that marine ecological management tools, which are normally applied to observations, can be applied to ocean model forecasts to give early warnings. They apply management tools for whale habitat area and loggerhead turtle avoidance in the Californian Coastal Ecosystem to a large ensemble of global ocean forecast systems to show that they can be skilfully used despite the low modelled spatial resolution. They also applied the same methods to a 3-member regional downscaling ensemble, to show the importance of regional downscaling, and a “reduced” global ensemble (the same the members of the global ensemble) so show the importance of resolution vs. ensemble size. They conclude that marine ecological management tools may be directly applied to available global forecast ensemble, to give meaningful ecological forecasts, and furthermore, that expensive regional downscaling may not be required – I will discuss this point below. They show that the regional model outperforms the global model on a case-by-case basis, but that this benefit is outweighed by the much larger global model ensemble size.

Their methodology, data analysis, and statistical methods of their assessment are sound, although I think the two example from the same region mean that care will be needed when considering other regions of the world. I think there is enough details to reproduce this methodology.

I think this is a good paper that raises awareness of the possibility and utilisation of marine ecological forecasts for marine management users. The authors have provided two useful examples, and I think it's important that some end users are not put off using global models for fear that they may be of too low resolution. However, in some places of the world,

global models do not include processes that dominate locally. I think it's important to add some nuance to their statement that dynamic downscaling may not be required, as for some regions global simulations are simply inadequate.

My expertise is in regional downscaling of ocean models for the northwest European shelf seas. We have a very broad continental shelf, and large tides, that control the local mixing regime, and so tend to dominate the local oceanography. Global ocean models do not currently simulate tides, so do not correctly represent stratification and mixing. Within our region, while some conditions can be simulated and forecast with global models, the exclusion of such an important process can make such forecasts useless for many parameters. Furthermore, missing processes can add a systematic bias to a global model, which could mean that all members of a global ensemble can be wrong – in which case a much smaller ensemble, or even a single realisation of a regional model can be more accurate. Extreme care must be made must be taken when using global forecast for our region, and other regions similar to the NW European Shelf Seas.

Global ocean models can adequately simulate the “near-coast” in other regions of the world oceans, particularly those with much smaller continental shelves. I gather the California Coastal System, and the (southern and eastern?) coast of Australia may be places where the direct use of global ocean models is appropriate, both regions where marine ecological forecast systems are established. I think the importance of downscaling in our region is one of the reasons why ecological forecasting is not as advanced or prevalent as in the US and Australia.

The two examples they provided in this paper are very good but are both in regions where shelf seas processes are perhaps less important. I think this paper should be accepted and I don't ask for additional examples in regions where shelf seas processes are important, but I think this needs to be reflected more strongly in the discussion. While the authors do say “Researchers should test the performance of available forecasts before assuming their resolution precludes a specific application” it suggests that some ecological processes and features are large scale and some are small scale, and this is the main consideration when choosing between global and regional models.

There are several papers that discuss the importance of regional downscaling in our (and other) regions, and we have even started to look at how applicable seasonal forecasts are to the NW European Shelf seas. These sort of papers could give some insight into the strengths and weaknesses of using global models directly. I can suggest some papers if that would be useful.

**REVIEWER COMMENTS**

Reviewer #1 (Remarks to the Author):

At a high level, this manuscript was methodologically sound, original, and provides a useful
contribution to the emerging scientific literature on ecological forecasting.

My primary comments on the paper are a few small places where I think methodological details
and figures could be made more clear:

Thank you for your constructive review. We have clarified our methods and figures, and believe
the manuscript is improved by these revisions. Please see below individual responses to your
comments.

* The paragraph on lines 291-298 is repetitive with lines 208-221.

We have revised the text and removed the repetition (ln 354-356). The section now reads:

*"We conducted a retrospective forecast for the Global model (73 ensemble members for 1981-*
*2020), as well as for the three forecast configurations that aimed to compare the performance of*
*global and downscaled forecasts (Figure 1; 1981-2010)."*

* On line 304 it's not clear how Monthly SST thresholds were set.

Line 332-334 details the methods for how thresholds were set, but in this section we have
added the threshold in parenthesis (ln 362). It now reads:

*"The monthly SST thresholds (mean monthly SST from the coast to 75 km offshore) used to*
*calculate the HCI were based on years 1981-2010."*

* On lines 329-330 the explanation of forecast accuracy and SEDI are too short and it's not
clear that equations for this will be presented later. Even when you get to it later, it's not clear
what the "several properties" are that make SEDI useful.

We had added a sentence to make it clear that more detail and equations for the metrics will be
described in the subsequent paragraphs (ln 404). We have also added additional text to specify
the properties that make SEDI useful (ln 447-451):

*"SEDI is a useful metric for assessing the skill of forecasting extreme events^{7,47}. It is non-*
*degenerate, meaning that it does not trend towards zero or infinity as rarity increases; it is not*
*influenced by changes in the frequency of events (known as base-rate independence); and it is*
*equitable, meaning that all forecasts including random forecasts receive the same expected*
*score (zero), irrespective of what method is used to generate random forecasts⁵⁰."*

* Line 348, the definition of a HCI event as any one that's below average seems like a really

lenient threshold, and it's not clear how this connect to the species biology or historically
observed bycatch.

Yes, this is a more lenient threshold than typically seen in the literature, but it is effective at
detecting anomalous conditions. That is, the long-term climatological signal is removed in the
calculation of the HCI so that values below the mean identify what is anomalously low relative to
the historical period. The HCI has been evaluated as an effective way to monitor ecosystem
shifts in coastal upwelling systems (e.g., changes in the spatial abundance and variability of krill,
anchovy, and biodiversity indices) and is currently used as a management tool for assessing
whale entanglement risk in the California Dungeness crab fishery (Santora et al., 2020;
Schroeder et al, 2022). The 50% threshold is currently used as a management tool and is one
of the reasons why we wanted to test its utility in a forecast configuration (as described in the
introduction). We have added a sentence to the methods to specify that the calculation of the
HCI is consistent with the existing management tool (In 422):

*"This calculation of the HCI is consistent with the existing management tool^{20,21}"*

Santora, J.A., Mantua, N.J., Schroeder, I.D., Field, J.C., Hazen, E.L., Bograd, S.J., Sydeman,
59 W.J., Wells, B.K., Calambokidis, J., Saez, L., et al. (2020). Habitat compression and ecosystem
shifts as potential links between marine heatwave and record whale entanglements. *Nat*
*Commun* 11, 536. 10.1038/s41467-019-14215-w.

Schroeder, I.D., Santora, J.A., Mantua, N., Field, J.C., Wells, B.K., Hazen, E.L., Jacox, M., and
Bograd, S.J. (2022). Habitat compression indices for monitoring ocean conditions and
ecosystem impacts within coastal upwelling systems. *Ecological Indicators* 144, 109520.
10.1016/j.ecolind.2022.109520.

* Line 352: also not completely clear how these thresholds were set. They do seem connected
to historically observed events, but its not clear why these specific thresholds were chosen (e.g.
was there some sort of logistic regression between SSTA and events?)

Likewise, the TOTAL tool is an existing management tool, and its threshold was chosen based
on a comprehensive analysis of sea turtle sightings relative to ocean temperature anomalies
(Welch et al., 2019). We have added a sentence explaining this (In 425):

*"This calculation of the TOTAL threshold is consistent with the existing management tool²²."*

Welch, H., Hazen, E.L., Briscoe, D.K., Bograd, S.J., Jacox, M.G., Eguchi, T., Benson, S.R.,
Fahy, C.C., Garfield, T., and Robinson, D. (2019). Environmental indicators to reduce
loggerhead turtle bycatch offshore of Southern California. *Ecological Indicators* 98, 657–664.

* Line 538: I'm not a fan of "available upon request" because there's a lot of evidence that such
data quickly becomes lost to the community. Is there a reason HCI and TOTAL in particular
can't be provided?

Thank you for flagging this and the encouragement to upload the results. We have uploaded our
forecasts to Dryad, uploaded our code to a public repo on github, and added source data for all

the figures. The real-time HCI and TOTAL tools are operational with the data freely available.
We have updated our data availability and code availability sections accordingly (ln 467-482),
but please note the dryad link won't work until the paper is published:

***"Data Availability***

*Global SST forecasts can be accessed at the North American Multi-Model Ensemble*
*(<https://www.cpc.ncep.noaa.gov/products/NMME/>). Output from the UCSC CCS reanalysis is*
*available from <https://oceanmodeling.ucsc.edu>. Downscaled SST forecasts, and forecasts of*
*HCI and TOTAL are available on dryad at <https://doi.org/10.5061/dryad.z08kprjr>. Source data*
*are provided with this paper. The real-time HCI*
*(https://oceanview.pfeg.noaa.gov/whale_indices/) and TOTAL*
*(<https://coastwatch.pfeg.noaa.gov/loggerheads/index.html>) management tools are publicly*
*accessible.*

***"Code Availability***

*Code used to make forecasts and figures is available on GitHub:*

*https://github.com/stephbrodie1/Ecological_Forecast*

* Figure 2: the use of dots in 2B to indicate significance creates the impression in 2C and 2D
that no values are significant. Furthermore, the threshold for significance is not 100% clear for
these metrics -- I wonder if you could use some sort of Monte Carlo approach (e.g.,
bootstrapping) to generate a significance test?

*This is a good idea, thanks for this suggestion. We have used bootstrapping to generate*
*random forecasts and test significance. We have updated the methods to describe in detail our*
*approach (ln 460-465), updated the results text (ln 92-97), and updated Figures 2 and 3.*

*Methods addition (ln 460-465):*

*"The significance of two skill metrics, forecast accuracy and SEDI, were quantified using*
*bootstrapping. For every month and lead time, we sampled random forecasts (as either a 1 or 0*
*to classify whether an event did or didn't occur) from the observed events for every year and*
*calculated forecast accuracy and SEDI. This was repeated 1000 times. The 95% confidence*
*intervals were calculated from the skill of the 1000 random forecasts, where significance was*
*defined as skill exceeding the 97.5 percentile of the random forecasts."*

*Results revisions (ln 92-97):*

*"For the HCI, forecasts of high compression events - which are associated with high whale*
*entanglement risk - were typically skilful from 0.5-1.5 months lead time across all months of the*
*year (Figure 2B-D). Significant forecast skill of high compression events extends out to 8.5*
*months for forecasts in February-March. Overall we see that forecast skill varies depending on*
*the metric of skill used (Figure 2B-D) but in general skill is higher in winter and spring, which are*

*important seasons for mitigating whale entanglement risk as they coincide with both fishing*
*activity and whale migrations (Figure 2B-D)."*

**Revised Figure 2**

* Figure 3, S1, S2: same. Would be nice to have a more clear visual indicator of significance
*As above.*

**Results revision (In 109):**
*"In our second case study, TOTAL closure conditions were skilfully forecast at 6.5 months lead*
*time, with significant skill extending out to 11.5 months by some metrics (Figure 3)"*

Revised Figure 3:

* Totally nit-picky, but I found the British spelling of "skilful" instead of "skillful" to be really
distracting

Thanks for sharing this but we are unsure of the best approach here. While the study is US-
based the journal is UK-based – so, like other forecasting papers published in the Nature suite
(e.g. Jacox et al., 2022; Payne et al., 2022), we have opted for the British spelling. I will defer to
the editor to make the best choice for the spelling.

Reviewer #2 (Remarks to the Author):

The authors show that marine ecological management tools, which are normally applied to
observations, can be applied to ocean model forecasts to give early warnings. They apply
management tools for whale habitat area and loggerhead turtle avoidance in the Californian
Coastal Ecosystem to a large ensemble of global ocean forecast systems to show that they can
be skilfully used despite the low modelled spatial resolution. They also applied the same methods
to a 3-member regional downscaling ensemble, to show the importance of regional
downscaling, and a “reduced” global ensemble (the same the members of the global ensemble)
so show the importance of resolution vs. ensemble size. They conclude that marine ecological
management tools may be directly applied to available global forecast ensemble, to give
meaningful ecological forecasts, and furthermore, that expensive regional downscaling may not
be required – I will discuss this point below. They show that the regional model outperforms the
global model on a case-by-case basis, but that this benefit is outweighed by the much larger
global model ensemble size.

Their methodology, data analysis, and statistical methods of their assessment are sound,

although I think the two example from the same region mean that care will be needed when
considering other regions of the world. I think there is enough details to reproduce this
methodology.

I think this is a good paper that raises awareness of the possibility and utilisation of marine
ecological forecasts for marine management users. The authors have provided two useful
examples, and I think it's important that some end users are not put off using global models for
fear that they may be of too low resolution. However, in some places of the world, global models
do not include processes that dominate locally. I think it's important to add some nuance to their
statement that dynamic downscaling may not be required, as for some regions global
simulations are simply inadequate.

My expertise is in regional downscaling of ocean models for the northwest European shelf seas.
We have a very broad continental shelf, and large tides, that control the local mixing regime,
and so tend to dominate the local oceanography. Global ocean models do not currently simulate
tides, so do not correctly represent stratification and mixing. Within our region, while some
conditions can be simulated and forecast with global models, the exclusion of such an important
process can make such forecasts useless for many parameters. Furthermore, missing
processes can add a systematic bias to a global model, which could mean that all members of a
global ensemble can be wrong – in which case a much smaller ensemble, or even a single
realisation of a regional model can be more accurate. Extreme care must be made must be
taken when using global forecast for our region, and other regions similar to the NW European
Shelf Seas.

Global ocean models can adequately simulate the “near-coast” in other regions of the world
oceans, particularly those with much smaller continental shelves. I gather the California Coastal
System, and the (southern and eastern?) coast of Australia may be places where the direct use
of global ocean models is appropriate, both regions where marine ecological forecast systems
are established. I think the importance of downscaling in our region is one of the reasons why
ecological forecasting is not as advanced or prevalent as in the US and Australia.

The two examples they provided in this paper are very good but are both in regions where shelf
sees processes are perhaps less important. I think this paper should be accepted and I don't
ask for additional examples in regions where shelf seas processes are important, but I think this
needs to be reflected more strongly in the discussion. While the authors do say “Researchers
should test the performance of available forecasts before assuming their resolution precludes a
specific application” it suggests that some ecological processes and features are large scale
and some are small scale, and this is the main consideration when choosing between global
and regional models.

There are several papers that discuss the importance of regional downscaling in our (and other)
regions, and we have even started to look at how applicable seasonal forecasts are to the NW
European Shelf seas. These sort of papers could give some insight into the strengths and
weaknesses of using global models directly. I can suggest some papers if that would be useful.

Thank you for your thoughtful and constructive review. Your insights and suggestions to expand the discussion have greatly improved the manuscript. I enjoyed reading your review and have several responses below – some of which summarize direct changes to the manuscript, and some are to continue the dialogue with you.

In response to your comments about expanding our discussion and explicitly considering shelf-processes, we have:

- Revised Figure 5 to more explicitly mention tides, shelf-process, and sub-surface structure as potential fine-scale features that downscaling can capture (see right column, 4th row).
- Added a sentence to the first discussion paragraph to flag that there are pros and cons to global and regional downscaled forecasts, of which we discuss in detail (Ln 160-162).
- Added additional text to suggest that downscaling will be particularly relevant for regions where fine-scale physical processes need to be captured, and added a citation (Tinker et al., 2020). Ln 231-234).
- Cited Stock et al. 2011 for additional insight on downscaling approaches and consideration of shelf-processes (Ln 215, citation #36).
- Added in some additional text to acknowledge the limitations of only using case studies from the CCS, as well as pointed to some research (Stock et al., 2015) that has shown the capacity of skillful global SST forecasts for coastal regions (including shelf seas). (Ln 237-242).

More broadly, we also wanted to share some more context about the CCS. The CCS does have a narrow shelf relative to your region of expertise, however there are many fine-scale and important processes in the CCS (e.g. coastal upwelling, coastal surface and subsurface currents, coastally trapped waves) that occur on scales of ~10 km and are not properly resolved by the global models. Our results thus showcase two useful examples of where global models provide skill despite the importance of fine-scale processes. Our discussion touches on the benefits of using spatially- and temporally coarse management case studies, and flags that tools that require more finer resolution may benefit from downscaling (Ln 229-234). Nonetheless, we believe it's important to convey that (1) increased model resolution does not necessarily generate forecast skill at that higher resolution; (2) downscaling is not always necessary to produce skillful forecasts; and (3) downscaling may not offer an improvement when all other factors (e.g. computational limitations) are taken into account.

In response to your comments on why the USA and Australia have more applications of ecological forecasts: It might be somewhat biased by where skillful forecasts occur, but also that more effort is concentrated in regions where leading/pioneering researchers are based and that these regions may have management protocols that more readily lend themselves to applications of forecasts. Stock et al., 2015 examines coastal ecosystems around the world and finds SST anomalies have significant forecast skill in a number of regions, including in NW Europe (see Fig 11 of Stock et al., 2015), and yet these non-European regions haven't had particular advances in ecological forecasting applications. Some of the grassroots efforts (e.g.

Ecological Forecasting Initiative) to increase education of ecological forecasting and scientific
capacity may help to offset some of these research biases as the field expands and moves
forward.

In response to your concerns about biases in the global model ensemble members rendering
forecasts useless: this depends on the application. For many applications, the bias does not
preclude use of the model if it can accurately reproduce variability. For example, if one wants to
predict extreme events (like when temperature will be especially warm or cold), a model can
identify those events even if its mean state is biased. In our approach, we calculated anomalies
relative to the model's own climatology to help remove the model bias. For the CCS, global
models tend to be too warm as they don't fully capture the strength of seasonal upwelling, but
by calculating anomalies in this way we can remove the influence of those biases to effectively
predict anomalies.

Finally, we would be receptive to seeing any references you suggest on these issues (as you
indicate above).

References:

Stock, C.A., Alexander, M.A., Bond, N.A., Brander, K.M., Cheung, W.W.L., Curchitser, E.N.,
Delworth, T.L., Dunne, J.P., Griffies, S.M., Haltuch, M.A., et al. (2011). On the use of IPCC-
class models to assess the impact of climate on Living Marine Resources. *Progress in*
*Oceanography* 88, 1–27. [10.1016/j.pocean.2010.09.001](https://doi.org/10.1016/j.pocean.2010.09.001).

Stock, C.A., Pegion, K., Vecchi, G.A., Alexander, M.A., Tommasi, D., Bond, N.A., Fratantoni,
P.S., Gudgel, R.G., Kristiansen, T., O'Brien, T.D., et al. (2015). Seasonal sea surface
temperature anomaly prediction for coastal ecosystems. *Progress in Oceanography* 137,
219–236. [10.1016/j.pocean.2015.06.007](https://doi.org/10.1016/j.pocean.2015.06.007).

Tinker, J., and Hermanson, L. (2021). Towards Winter Seasonal Predictability of the North West
European Shelf Seas. *Frontiers in Marine Science* 8.

REVIEWERS' COMMENTS

Reviewer #1 (Remarks to the Author):

I was generally happy with the first version of this manuscript and was excited to see the extent to which the authors adopted the few suggestions I provided, in particular the effort they put into bootstrapping significance tests for two additional analyses and their decision to make their code and data publicly available.

The only critique I have is very minor, and definitely not worth another round of review, and in bringing it up I openly acknowledge the authors face tough limits on paper length so they may not be able to do much to respond. That said, in response to my questions about the lack of clarity in where HCI and TOTAL come from, the authors provided very clear and compelling responses in the rebuttal doc, but then in the paper just wrote "This calculation of the [HCI,TOTAL threshold] is consistent with the existing management tool". I'd love to see just a few additional words in the paper itself to pull in a bit more context. For example, for TOTAL you could just change "is consistent" to "is [based on/calibrated against] sea turtle sighting data and consistent". For HCI you might use your rebuttal text "is used as a management tool for assessing whale entanglement risk in the California Dungeness crab fishery" instead of the uninformative/generic "is consistent with the existing management tool"

Reviewer #2 (Remarks to the Author):

Hi Stephanie,

Thanks for your response to my review, I was happy that you have thoroughly addressed my points and concerns, and I'm happy for the paper to be published.

I have not considered the other review, but I am sure that is in hand too.

Congratulations on a great paper!

Kind Regards

Jonathan

**REVIEWERS' COMMENTS**

Reviewer #1 (Remarks to the Author):

I was generally happy with the first version of this manuscript and was excited to see the extent
to which the authors adopted the few suggestions I provided, in particular the effort they put into
bootstrapping significance tests for two additional analyses and their decision to make their
code and data publicly available.

The only critique I have is very minor, and definitely not worth another round of review, and in
bringing it up I openly acknowledge the authors face tough limits on paper length so they may
not be able to do much to respond. That said, in response to my questions about the lack of
clarity in where HCI and TOTAL come from, the authors provided very clear and compelling
responses in the rebuttal doc, but then in the paper just wrote "This calculation of the
[HCI,TOTAL threshold] is consistent with the existing management tool". I'd love to see just a
few additional words in the paper itself to pull in a bit more context. For example, for TOTAL you
could just change "is consistent" to "is [based on/calibrated against] sea turtle sighting data and
consistent". For HCI you might use your rebuttal text "is used as a management tool for
assessing whale entanglement risk in the California Dungeness crab fishery" instead of the
uninformative/generic "is consistent with the existing management tool"

Thank you for your positive response to our revisions. We apologize we did not adequately
address your comment about identifying the thresholds for the management tools. We have
revised the manuscript to now state (In 356-362):

*"This calculation of the HCI is consistent with the existing management tool for assessing whale
entanglement risk in the California Dungeness crab fishery."*

*"This calculation of the TOTAL threshold is informed by sea turtle sightings data and is
consistent with the existing management tool for assessing bycatch risk in the Swordfish drift
gillnet fishery"*

Reviewer #2 (Remarks to the Author):

Hi Stephanie,

Thanks for your response to my review, I was happy that you have thoroughly addressed my
points and concerns, and I'm happy for the paper to be published.

I have not considered the other review, but I am sure that is in hand too.

Congratulations on a great paper!

Kind Regards

Jonathan

Thank you for reviewing and considering our revised manuscript.